# Peri-Procedural Troponin Elevation after Percutaneous Coronary Intervention for Left Main Coronary Artery Disease

**DOI:** 10.3390/jcm12010244

**Published:** 2022-12-28

**Authors:** Wojciech Jan Skorupski, Marta Kałużna-Oleksy, Przemysław Mitkowski, Włodzimierz Skorupski, Stefan Grajek, Małgorzata Pyda, Aleksander Araszkiewicz, Maciej Lesiak, Marek Grygier

**Affiliations:** 1st Department of Cardiology, Poznan University of Medical Sciences, 61-848 Poznan, Poland

**Keywords:** coronary artery disease, left main, percutaneous coronary intervention, periprocedural myocardial injury, troponin elevation

## Abstract

Left main (LM) percutaneous coronary interventions (PCI) are challenging and highly invasive procedures. Periprocedural myocardial injury (Troponin (Tn) elevation > 99th percentile) is frequently detected after LM PCI, being identified even in up to 67% of patients. However, the prognostic implications of periprocedural Tn elevation after LM PCI remain controversial. We aim to assess the impact and prognostic significance of the periprocedural troponin elevation on long-term outcomes in patients undergoing LM PCI in a real-world setting. Consecutive 673 patients who underwent LM PCI in our department between January 2015 to February 2021 were included in a prospective registry. The first group consisted of 323 patients with major cardiac Troponin I elevation defined as an elevation of Tn values > 5× the 99th percentile in patients with normal baseline values or post-procedure Tn rise by >20% in patients with elevated pre-procedure Tn in whom the Tn level was stable or falling (based on the fourth universal definition of myocardial infarction). The second group consisted of patients without major cardiac Troponin I elevation. Seven-year long-term all-cause mortality was not higher in the group with major Tn elevation (36.9% vs. 40.6%; *p* = 0.818). Naturally, periprocedural myocardial infarction was diagnosed only in patients from groups with major Tn elevation (4.9% of all patients). In-hospital death and other periprocedural complications did not differ significantly between the two study groups. The adjusted HRs for mortality post-PCI in patients with a periprocedural myocardial infarction were not significant. Long-term mortality subanalysis for the group with criteria for cardiac procedural myocardial injury showed no significant differences (39.5% vs. 38.8%; *p* = 0.997). The occurrence of Tn elevation (>1×; >5×; >35× and >70× URL) after LM PCI was not associated with adverse long-term outcomes. The results of the study suggest that the isolated periprocedural troponin elevation is not clinically significant.

## 1. Introduction

The left main coronary artery (LM) is crucial for maintaining blood flow to most of the left ventricular (LV) myocardium, therefore significant LM narrowing puts this cohort at great risk [1,2]. Decisions regarding the choice of the revascularization method i.e., percutaneous coronary interventions (PCI) vs. coronary artery bypass graft (CABG) remain in the heart team’s tasks and are individualized for each patient. Advances in PCI, such as adequate patient selection, development in device technology, stenting techniques, enhanced interventional cardiologist experience, and modern post-procedure therapy, have made PCI a safe and efficient alternative to CABG surgery for LM coronary artery disease [3,4,5]. Despite the great efficiency and high safety, it should be emphasized that LM PCI are challenging and highly invasive procedure. Periprocedural myocardial injury (Troponin (Tn) > 99th percentile) is frequently discovered after LM PCI, being recognized even up to 67% of patients using a standard Tn assay [6].

Myocardial infarction (MI) is a frequently utilized endpoint in cardiology scientific studies, indicating irreversible myocardial necrosis mostly due to acute myocardial ischemia [7]. There are various types of MI with different mechanisms involved. Periprocedural myocardial infarction (PMI) refers to MI that arises during PCI or CABG. Interventional cardiologists, under the patronage of the Society for Cardiac Angiography and Intervention (SCAI), generally concurred on a definition of a clinically significant MI based on an increase of creatine kinase-myocardial band (CK-MB) >10 times or troponin >70 times the upper limit of normal (ULN) as an independent criterion [8]. Nevertheless, upon consideration, the specialists developing the latest Fourth Universal Definition of MI (UDMI) have decided to preserve the previous (Third UDMI) definition. Fourth UDMI defines percutaneous coronary intervention-related MI (MI type 4a) as an elevation of cardiac troponin (Tn) values more than five times the 99th percentile upper reference limit (URL) in patients with normal baseline values [9]. In patients with elevated pre-procedure Tn in whom the Tn level is stable (≤20% variation) or falling, the post-procedure Tn must rise by >20% [9]. However, the absolute post-procedural value must still be at least five times the 99th percentile ULN [9]. The prognostic implications of significant periprocedural Tn elevation after percutaneous coronary intervention are still controversial [10,11]. In addition, a lower troponin elevation (>5 times the 99th percentile) in PCI compared to CABG (>10× ULN) is required to diagnose peri-operative myocardial infarction [9]. No established agreement amongst cardiology societies and scientific groups has led to a variety of definitions with different cardiac biomarkers (CK-MB or Tn), deliberations on whether the multiplicity of ULN should change after PCI or CABG, and whether clinical characteristics such as electrocardiogram changes, new artery occlusions, new regional wall kinetic abnormalities, or loss of viable myocardium should be considered [4,8,9,12,13,14,15,16,17,18]. Most importantly, the influence of PMIs definition on mortality is various. Since an appropriate definition of PMI has not been established, and patients’ survival differs depending on the different definitions used, perhaps PMI should be removed from primary composite endpoints [19]. Therefore, the optimal PMI definition and the consequences of this selection on the results should be further explored. A summary of all PMI definitions is provided in Table 1.

We aim to assess the impact and prognostic significance of the periprocedural troponin elevation on long-term results in patients undergoing LM PCI in a real-world setting.

## 2. Materials and Methods

This manuscript is a part of a series of scientific articles on LM disease. The design of the registry has been previously reported [20,21,22]. In brief, subsequent 673 LM PCI patients from the period between January 2015–February 2021 were included in the study. Patients with significant LM stenosis (≥50% diameter) engaging or not the ostial narrowing of the left anterior descending artery (LAD), circumflex coronary artery (LCx), or both were enrolled in the study. Moderate lesions were assessed with the use of intravascular ultrasound imaging (IVUS) (minimal lumen area of 6.0 mm for LM). Terminal patients (life expectancy below one year) were eliminated from the further analysis. All PCI interventions were performed by an experienced invasive cardiologist after the heart team sat with a cardiac surgeon. 

LM patients were categorized into two groups. The first group was composed of 323 patients with major cardiac Troponin I elevation defined as an elevation of Tn values more than 5 times the 99th percentile URL in patients with normal baseline values or post-procedure Tn rise by >20% in patients with elevated pre-procedure Tn in whom the Tn level was stable or falling (however, the absolute post-procedural value still needs to be at least 5 times the 99th percentile URL). Tn elevation cut-off values were determined based on the criteria for PCI-associated MI ≤ 48 h after the PCI procedure (MI type 4a) from the most recent Fourth UDMI [9]. Patients at our facility had Troponin I levels measured prior to and after the PCI intervention as part of the routine clinical management. The second group consisted of patients without major cardiac Troponin I elevation (Figure 1).

Baseline clinical and angiographic characteristics, procedural information, in-hospital and follow-up outcome data were examined. Contrast-induced nephropathy was determined as the deterioration of kidney function, depicted as either a 25% or ≥0.5 mg/dL (44 μmol/L) serum creatinine elevation from baseline level within 48 h [23]. Chronic kidney disease was determined using the Cockcroft–Gault equation [24]. All coronary artery bifurcations were categorized using the Medina classification [25]. In-hospital death, in-hospital MI, and long-term all-cause death were established as a primary outcome. Secondary outcome of the study was Tn analysis according to various cutoffs of postprocedural Troponin levels. 

Periprocedural MI type 4a diagnosis, in addition to Tn elevation, required proof of new myocardial ischaemia (i.e., new ECG abnormalities, cardiac imaging evidence, or PCI complications associated with reduction in coronary artery blood flow [9]. IVUS or optical coherence tomography (OCT) methods were used in 31.9% of patients and were not investigated thoroughly in this paper. The mean and median follow-up for the cohort were respectively 1385 and 1411 days (interquartile range: 938 days, max: 2553 days). Survival analysis data were gathered by telephone contact or with the use of National Health Fund documentation. 

All analyses were performed with STATISTICA 13.7 (StatSoft, Inc., Tulsa, OK, USA). Continuous variables were shown as means ± SD or medians (IQR). Categorical variables were summarized as counts or percentages and were compared between groups using the test for proportions. The normality distribution was analyzed using the Shapiro–Wilk test. Differences between continuous variables were tested with Student’s *t*-test or nonparametric Mann-Whitney test. The survival probability at follow-up was calculated using the Kaplan–Meier method. Log-rank tests were used to compare survival between the different groups. A two-sided *p* < 0.05 was considered statistically significant.

## 3. Results

A total of 673 consecutive LM PCI patients (mean age: 68.8 ± 9.2 years, 76.4% males) were divided into two groups according to cardiac Troponin I elevation as defined above. The first group was composed of 323 patients with major cardiac Troponin I elevation and the second group consisted of patients without major cardiac Troponin I elevation. Patients’ baseline clinical characteristics are presented in Table 2. There were no significant differences in the incidence of major cardiovascular risk factors. On admission, stable coronary artery disease was found in a smaller percentage in the group with major Tn elevation (51.7% vs. 62.3%; *p* = 0.006), the tendency towards the higher occurrence of unstable CAD (27.2% vs. 21.4%; *p* = 0.078) and NSTEMI (17.0% vs. 12.3%; *p* = 0.081) was visible. Prior PCI LAD was found to be more frequent in groups without Tn elevation (19.5% vs. 26.6%; *p* = 0.030). Left ventricular ejection fraction (LVEF) (49.1 ± 11.4% vs. 50.4 ± 11.3%; *p* = 0.150) and left ventricular end-diastolic diameter (LVEDD) (52.0 ± 7.1 mm vs. 51.1 ± 7.3 mm; *p* = 0.151) was similar in both groups. 

Baseline angiographic characteristics are shown in Table 3. Localization of the LM lesions was similar in both groups. Not-ostial LAD (57.6% vs. 50.3%; *p* = 0.058) and LCX (41.5% vs. 34.6%; *p* = 0.065) lesions were found more often in groups with major Tn elevation, however, this difference was not significant. The trend toward a higher frequency of LM plus two-vessel disease (30.7% vs. 24.9%; *p* = 0.093) and LM plus three-vessel disease (14.2% vs. 10.0%; *p* = 0.091) was observed. There was no significant difference in the frequency of Medina bifurcation types between both groups. Syntax Score was significantly higher in patients with major Tn elevation (26.2 ± 10.2 vs. 24.3 ± 9.9; *p* = 0.011). The EuroScore II value was non-significantly higher in Group with Tn elevation (2.35 ± 1.99 vs. 2.24 ± 1.83; *p* = 0.788).

Procedural characteristics are presented in Table 4. A procedure success ratio was high in both patient fractions. The number of stents (1.9 ± 0.9 vs. 1.7 ± 0.8; *p* < 0.001) and the total length (45.1 ± 24.9 mm vs. 37.5 ± 21.8 mm; *p* < 0.001) were notably higher in the group with major Tn elevation. Periprocedural fluoroscopy time did not differ significantly, however, a dose of radiation was significantly higher in the group with Tn elevation (1500 ± 896 vs. 1317 ± 822; *p* = 0.010). Artery access did not differ in both groups, and radial access was used more frequently. Only second-generation drug-eluting stents were used in this study. All stenting techniques are summarized in Table 4. Two-stent techniques were performed more frequently in groups with major Tn elevation (27.2% vs. 19.4%; *p* = 0.016). The crushing technique constituted 38.5% of two-stent techniques.

Procedural clinical results are summarized at the end of Table 4. Naturally, PMI type 4a was diagnosed only in patients from group with major Tn elevation (4.9% of all patients). Contrast induced nephropathy was found more frequently in the group with major Tn elevation (5.6% vs. 3.1%; *p* = 0.121), but the difference was not significant. Frequency of other periprocedural complications were on the similar levels in two groups. There was no significant difference in the seven-year long-term all-cause mortality between groups with and without major Tn elevation (36.9% vs. 40.6%; *p* = 0.818) (Figure 2).

### 3.1. Subanalysis: Patients with Periprocedural Myocardial Infarction

At seven years, the adjusted HRs for mortality after LM PCI in patients with a periprocedural myocardial infarction were not significant (21.2% vs. 25.5%; HR: 0.833; 95% CI: 0.426 to 1.629; *p* = 0.593).

### 3.2. Subanalysis: Patients with Criteria for Cardiac Procedural Myocardial Injury

In a subanalysis, we decided to examine the criteria for cardiac procedural myocardial injury. Cardiac procedural myocardial injury is arbitrarily defined by increases in Tn values (>99th percentile URL) in patients with normal baseline values (≤99th percentile URL) or a rise of Tn values > 20% of the baseline value when it is above the 99th percentile URL but it is stable or falling [9]. In terms of troponin elevation, the difference is that the criteria for cardiac procedural myocardial injury elevation do not require a five-fold increase compared to the 99th percentile URL (in comparison to periprocedural MI criteria). In our group, the criteria for cardiac procedural myocardial injury were met in 56.0% of patients. Long-term mortality in that subanalysis showed no significant differences (39.5% vs. 38.8%; *p* = 0.997) between the groups (Figure 3).

### 3.3. Subanalysis: Different Postprocedural Troponin Levels Cutoffs

Moreover, the graph presenting Kaplan-Meier curves for all-cause death according to different cutoffs of postprocedural Tn levels is provided. There is no significant difference in long-term mortality although the highest mortality in patients with >70× Tn elevation is visible (49.5%; *p* = 0.142) (Figure 4).

### 3.4. Subanalysis: Different Postprocedural CK-MB Mass Levels Cutoffs

Forty-eight percent (323 patients) of patients from our group had CK-MB mass levels measured before and after the LM PCI procedure. Additionally, we decided to investigate the association of CK-MB mass with long-term all-cause mortality in a subanalysis. The Kaplan-Meier curves for all-cause mortality according to different cutoffs of postprocedural CK-MB levels are provided. There is no significant difference in long-term mortality (*p* = 0.560) (Figure 5).

## 4. Discussion

Clinically relevant PMI should be determined as those leading to major cardiovascular events, especially death. Periprocedural myocardial injury is frequently detected after LM PCI, recognized in the present study in over 50% of patients using a standard Tn assay. Nevertheless, the threshold for post-PCI biomarker elevations representing clinically significant periprocedural myocardial injury in LM PCI patients remains controversial. In recent years, this topic has been gaining more and more attention [6,26].

To the best of our knowledge, based on the literature review, this is the first paper to compare the groups of patients with significant Tn I elevation after LM PCI with patients without post-procedure Tn I elevation in a real-world setting. The aim of the present study was to assess the impact and prognostic significance of the periprocedural troponin elevation on long-term outcomes in patients undergoing PCI of LM CAD.

In this paper, groups were created based on Tn elevation values more than five times the 99th percentile URL in patients with normal baseline values or post-procedure Tn rise by >20% in patients with elevated pre-procedure Tn. These categories were used as the basal groups in tables, because these criteria are believed to have the greatest clinical implications and are used in clinical practice more often than the criteria for cardiac procedural myocardial injury. The main findings of this real-life large study examining the prognostic implications of periprocedural myocardial injury and troponin elevations in more than 600 consecutive patients undergoing LM PCI is that the (1) occurrence of Tn elevation (>1× URL (minor); >5× URL (major); >35× URL and >70× URL) after LM PCI is not associated with adverse long-term outcomes and does not affect the prognosis, (2) LM PCI is an invasive procedure and the troponin increase occurs in more than 50% of PCI LM patients population, (3) PMI according to fourth UDMI following LM PCI are not associated with an increase in all-cause mortality at seven years, (4) a subanalysis of 323 patients showed that the occurrence of CK-MB elevation (>1× URL; >3× URL; >5× URL and >10× URL) after LM PCI is also not associated with adverse long-term outcomes.

Whether isolated elevation of Troponin level after PCI delivers long-term prognostic information data about mortality is an important issue. Circulating troponin after PCI composed of two groups: a pre-PCI or baseline fraction and a fraction that corresponds to PCI-associated Tn increase. Continuous microscopic cardiomyocyte loss during everyday life and cardiomyocyte renewal are two processes that contribute to physiological baseline circulating troponin levels [27,28]. The articles describing the effects of post-PCI biomarkers elevation on major adverse cardiac effects and mortality differ in terms of the biomarkers assessed, their influence, and cut-off values [29,30,31,32,33]. Some of these articles stated that the Tn elevation was not associated with increased mortality and did not offer prognostic data beyond that provided by the baseline level of the biomarker [18,34,35]. Therefore, the significance of change and factors that predispose to an elevation of post-PCI Tn levels after LM PCI remain unclear.

Recent analysis from the EXCEL trial stated that both peak CK-MB ≥ 10× URL and Tn ≥ 70× URL were independently predictive of five-year all-cause and cardiovascular mortality [14,26]. In the Excel study, confusion is associated with the PMI definition: PMI by study protocol definition correlates with increased 5-year all-cause and cardiovascular mortality after PCI and CABG, but the fourth UDMI definition PMI is related only to increased mortality in the CABG group [14]. Similar results were obtained, in the SYNTAX where CK-MB ≥ 10× URL post-PCI was related to 30-day and a 10-year increase in all-cause mortality, SYNTAX trial did not report troponin levels analysis [13]. In the study by Hao-Yu et al., only post-procedural CK-MB levels elevations: ≥3× and ≥10× URL independently predicted increased 3-year cardiovascular and all-cause mortality, whereas different Tn I cutoff levels did not show such correlations [6]. Moreover, only the SCAI definition of PMI (but not ARC-2 and Fourth UDMI) was associated with higher cardiovascular (HR: 4.93; 95% CI: 1.92–12.69) and all-cause (HR: 3.11; 95% CI: 1.33–7.27) mortality after LM PCI [6]. The influence of differing definitions of PMI was also investigated in the ISCHEMIA study, wherein a primary definition used CK-MB and a second one used Tn. PMIs were not related to all-cause death (HR: 1.14 [95% CI: 0.42–3.08] and 1.06 [95% CI: 0.56–2.02]) or cardiovascular death (HR: 1.99 [95% CI: 0.73–5.43] and 1.24 [95% CI: 0.57–2.68]) using either definition [16], ISCHEMIA had a median follow-up of 3.2 years.

In our study, unlike most of the trials above, none of the Tn elevation cutoffs was related to the increase in the seven-year all-cause mortality (*p* = 0.142). An identical situation, with no effect on long-term death rates, occurs in terms of CK-MB elevations (*p* = 0.560). Moreover, the adjusted HRs for the seven-year all-cause mortality post-PCI in patients with a PMI by Fourth UDMI were not significant (HR: 0.833; 95% CI: 0.426 to 1.629; *p* = 0.593). PMI type 4a in our study was found in 4.9% of all patients, these results are in line with other real-world LM studies, in the DELTA 2 registry PMI was diagnosed in 4.0% of patients who underwent LM PCI [36]. It should be noted that the Tn elevation by Fourth UDMI for PMI was met in 48% of patients and the criteria for periprocedural myocardial injury in 56% of the patients. Earlier trials exposed that the three-year mortality in patients with unprotected LM CAD who received drug-only treatment was up to 50% [37,38,39], post-procedure elevation in Tn levels should not come as a surprise after life-saving LM PCI.

In patients with major Tn I elevation Syntax score, a number of implanted stents, radiation time, and radiation dose were higher and two-stent techniques were used more often. Therefore, it can be assumed that the PCI procedures performed on these patients were more complex. However, the procedure success ratio was high and similar in both study groups. It should be emphasized that patients qualified for PCI of LM often have severe comorbidities that disqualify them from CABG. For these patients, LM PCI is the only option, and the question that should be asked is whether it is possible to perform PCI on patients who could potentially undergo CABG without a periprocedural rise in Tn levels. Especially, that diagnosis of CABG-related MI (type 5 MI) requires a greater rise of Tn levels (>10× URL) than PCI-related MI type 4a (>5× URL) [9]. 

### Study Limitations

The presented article analyzes a real-life group of LM PCI interventions. One of the main constraints was the absence of a group that underwent a cardiac surgical procedure. Nevertheless, comparing that group to the coronary artery bypass graft cohort was from the beginning beyond the primary purpose of the analysis. Second, this research was conducted in a world setting, and LM disease is a highly diversified state. Therefore, the localization of the disease (ostial/shaft/bifurcation), the complexity of the coronary lesion, and coronary artery disease in distal branches could impact the results. Third, despite the introduced trial being a prospective registry, not entire clinically significant information was accessible. Next, all-cause mortality was evaluated based on the long-term outcomes, and a cardiovascular vs. non-cardiovascular death rates analysis could not be provided.

## 5. Conclusions

Elevated troponin concentrations after LM PCI in real-world settings are frequently observed. However, these findings are not associated with an increased risk of in-hospital death and all-cause mortality in a long-term follow-up. The results of the study suggest that the isolated periprocedural troponin elevation is not clinically significant. Moreover, the performed subanalysis implies no clinical significance of CK-MB elevation.

## Figures and Tables

**Figure 1 jcm-12-00244-f001:**
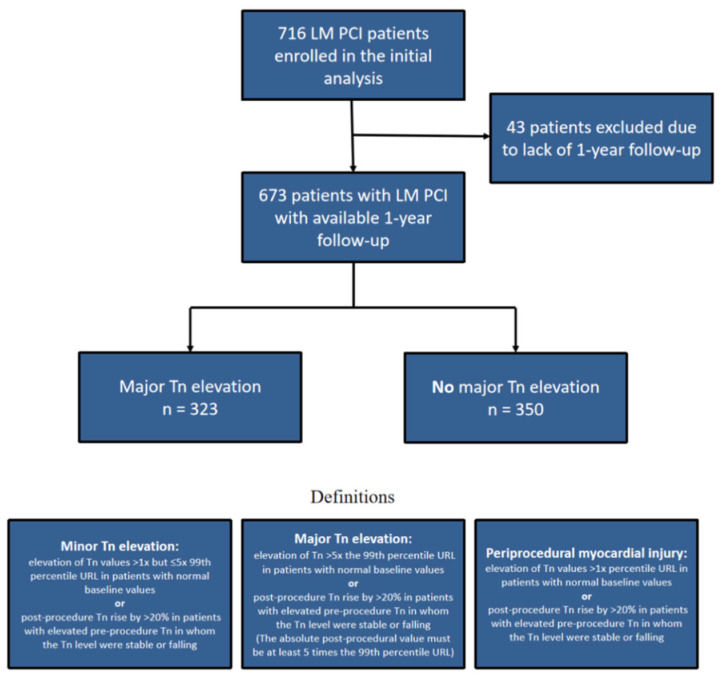
Flowchart presenting population size and definitions.

**Figure 2 jcm-12-00244-f002:**
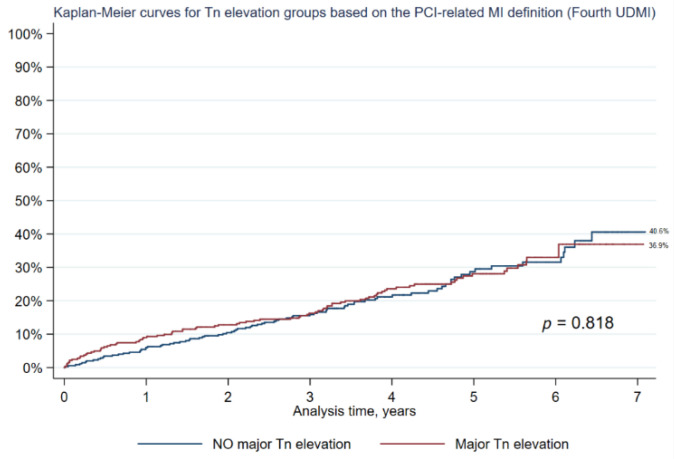
Kaplan-Meier analysis of all-cause mortality: major Tn elevation vs. NO major Tn elevation (based on PCI-related myocardial infarction definition from the fourth universal definition of myocardial infarction).

**Figure 3 jcm-12-00244-f003:**
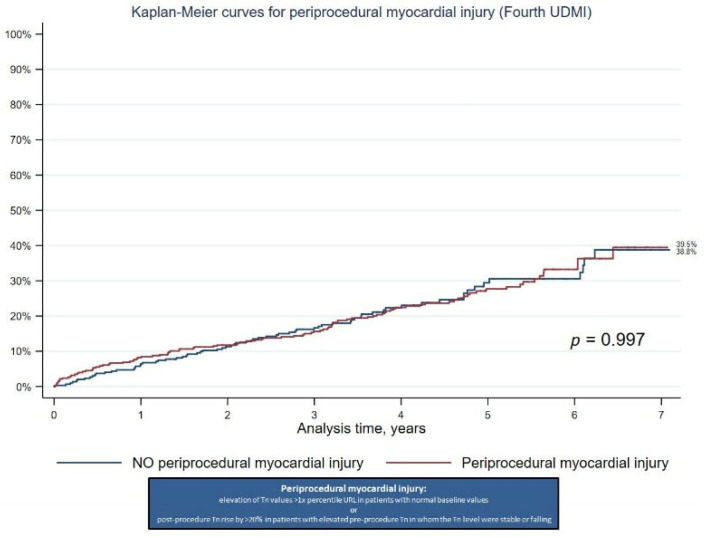
Kaplan-Meier analysis of all-cause mortality: Periprocedural myocardial injury vs. NO periprocedural myocardial injury (Fourth Universal Definition of Myocardial Infarction).

**Figure 4 jcm-12-00244-f004:**
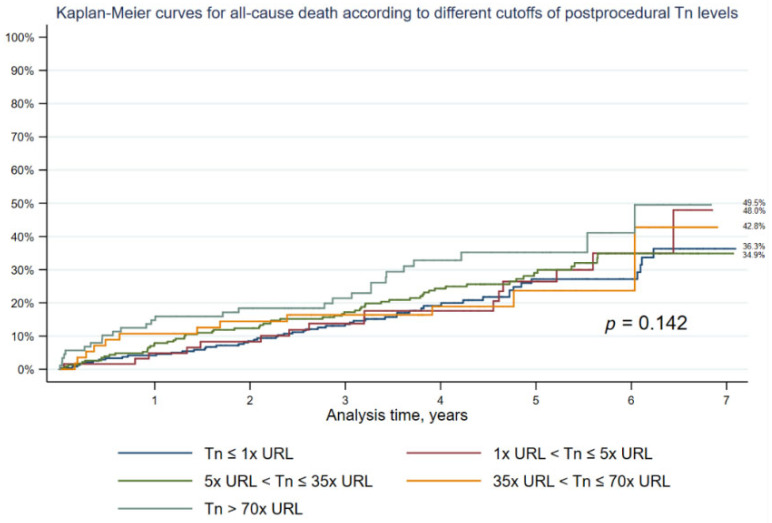
Kaplan-Meier curves for all-cause mortality according to different cutoffs of postprocedural Troponin levels.

**Figure 5 jcm-12-00244-f005:**
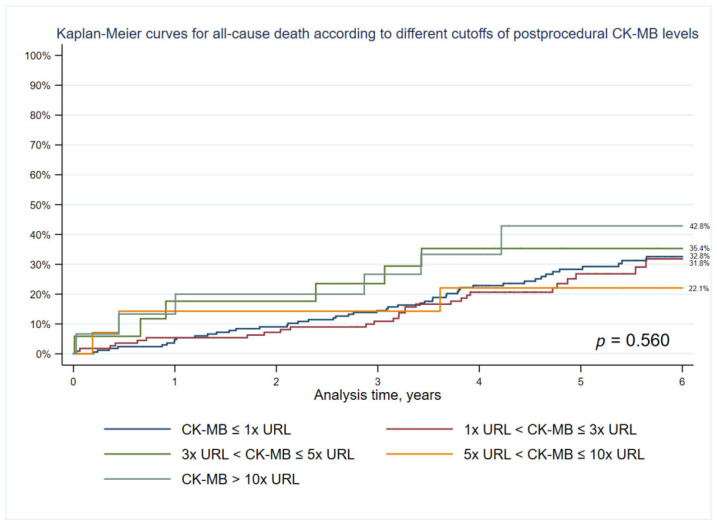
Kaplan-Meier curves for all-cause mortality according to different cutoffs of postprocedural CK-MB mass levels.

**Table 1 jcm-12-00244-t001:** Periprocedural myocardial infarction definitions.

PMI Definition	Biomarker Criteria	Additional Evidence Required
Fourth UDMI	1. Tn > 5× 99th percentile URL or 2. in patients with elevated pre-procedure Tn, post-procedure Tn must rise by >20%	1. New ischemic ECG changes or new Q waves 2. Angiographic findings 3. Imaging evidence: new loss of viable myocardium or new regional wall motion abnormality
ARC-2	1. Tn ≥ 35× URL with additional criteria or 2. Tn ≥ 70× URL with no additional evidence	1. New significant Q waves or equivalent 2. Angiographic findings 3. Imaging evidence: new loss of viable myocardium or new regional wall motion abnormality
SCAI	1. CK-MB ≥ 5× ULN with additional criteria or 2. CK-MB ≥ 10× ULN with no additional evidence	1. New Q waves or new persistent LBBB
EXCEL	1. CK-MB > 5× ULN with additional criteria or 2. CK-MB > 10× ULN with no additional evidence	1. New Q-waves or new persistent LBBB, 2. angiographic findings, 3. Imaging evidence: new loss of viable myocardium or new regional wall motion abnormality
SYNTAX	1. CK-MB/Total CK > 10% with additional criteria or 2. CK-MB > 5× ULN with additional criteria	1. Development of new pathological Q-waves
PRECOMBAT	1. CK-MB/Total CK > 10% with additional criteria or 2. CK-MB > 5× ULN with additional criteria	1. Development of new pathological Q-waves or new LBBB
ISCHEMIA	1. CK-MB > 5× ULN with additional criteria or 2. CK-MB > 10× ULN with no additional evidence	1. ST segment elevation or depression, new Q-waves, or new persistent LBBB, 2. Angiographic findings.

Fourth UDMI—fourth universal definition of myocardial infarction, Tn—troponin, LBBB—left bundle branch block.

**Table 2 jcm-12-00244-t002:** Baseline characteristics by treatment group.

Variable	Total *n* = 673	Major Troponin Elevation *n* = 323	NO Major Troponin Elevation *n* = 350	*p*-Value (Group 1 vs. Group 2)
Age (y)	68.8 ± 9.2	69.2 ± 9.1	68.4 ± 9.2	0.437
Gender (female)	159 (23.6%)	77 (23.8%)	82 (23.4%)	0.900
BMI (kg/m^2^)	28.1 ± 4.6	27.9 ± 4.6	28.3 ± 4.7	0.443
Hypertension	557 (82.8%)	268 (83.0%)	289 (82.6%)	0.891
Hyperlipidemia	351 (52.2%)	165 (51.1%)	186 (53.1%)	0.593
CKD	222 (33.0%)	109 (33.7%)	113 (32.3%)	0.687
DM	258 (38.3%)	122 (37.8%)	136 (38.9%)	0.772
Stroke/TIA	60 (8.9%)	25 (7.7%)	35 (10.0%)	0.303
COPD	55 (8.2%)	30 (9.3%)	25 (7.1%)	0.310
PVD	108 (16.0%)	54 (16.7%)	54 (15.4%)	0.649
AF	97 (14.4%)	51 (15.8%)	46 (13.1%)	0.329
Smoking (current)	248 (36.8%)	128 (39.6%)	120 (34.3%)	0.151
Prior MI	333 (49.5%)	165 (51.1%)	168 (48.0%)	0.424
Stable CAD	385 (57.2%)	167 (51.7%)	218 (62.3%)	0.006
Unstable CAD	163 (24.2%)	88 (27.2%)	75 (21.4%)	0.078
NSTEMI	98 (14.6%)	55 (17.0%)	43 (12.3%)	0.081
STEMI	23 (3.4%)	10 (3.1%)	13 (3.7%)	0.659
Prior PCI LAD	156 (23.2%)	63 (19.5%)	93 (26.6%)	0.030
Prior PCI LCX	108 (16.0%)	50 (15.5%)	58 (16.6%)	0.700
Prior PCI RCA	202 (30.0%)	97 (30.0%)	105 (30.0%)	0.993
Prior CABG	144 (21.4%)	71 (22.0%)	73 (20.9%)	0.722
LVEDD (mm)	51.6 ± 7.2	52.0 ± 7.1	51.1 ± 7.3	0.151
LVEF (%)	49.7 ± 11.4	49.1 ± 11.4	50.4 ± 11.3	0.150
EuroScore II	2.30 ± 1.91	2.35 ± 1.99	2.24 ± 1.83	0.788
Syntax Score	25.2 ± 10.1	26.2 ± 10.2	24.3 ± 9.9	0.011
0–22 (low)	290 (43.1%)	124 (42.8%)	166 (57.2%)	0.018
23–32 (intermediate)	225 (33.4%)	110 (48.9%)	115 (51.1%)	0.742
≥33 (high)	158 (23.5%)	87 (55.1%)	71 (44.9%)	0.042

BMI—body mass index, CKD—chronic kidney disease, DM—diabetes mellitus, TIA—transient ischemic attack, COPD—chronic obstructive pulmonary disease, PVD—peripheral vascular disease, AF—atrial fibrillation, MI—myocardial infarction, CAD—coronary artery disease, PCI—percutaneous coronary intervention, LAD—left anterior descending artery, LCX—left circumflex, RCA—right coronary artery, CABG—coronary artery bypass Graft, LVEDD—left ventricular end-diastolic diameter, LVEF—left ventricle ejection fraction.

**Table 3 jcm-12-00244-t003:** Baseline angiographic characteristics.

Variable	Total *n* = 673	Major Troponin Elevation *n* = 323	NO Major Troponin Elevation *n* = 350	*p*-Value (Group 1 vs. Group 2)
LM distal	540 (80.2%)	261 (80.8%)	279 (79.7%)	0.723
LM bifurcation	449 (66.7%)	212 (65.6%)	237 (67.7%)	0.567
LM trifurcation	85 (12.6%)	38 (11.8%)	47 (13.4%)	0.516
LM calcification	103 (15.3%)	52 (16.1%)	51 (14.6%)	0.582
LAD disease (not ostial)	362 (53.8%)	186 (57.6%)	176 (50.3%)	0.058
LCX disease (not ostial)	255 (37.9%)	134 (41.5%)	121 (34.6%)	0.065
Protected LM	100 (14.9%)	47 (14.6%)	53 (15.1%)	0.829
RCA recessive (a)	52 (7.7%)	24 (7.4%)	28 (8.0%)	0.782
RCA with critical stenosis (b)	81 (12.0%)	40 (12.4%)	41 (11.7%)	0.790
RCA total occlusion (c)	119 (17.7%)	62 (19.2%)	57 (16.3%)	0.323
Lack of RCA support to LMCAD (a + b + c)	252 (37.4%)	126 (39.0%)	126 (36.0%)	0.293
Extent of diseased vessels				
LM only	183 (27.2%)	82 (25.4%)	101 (28.9%)	0.312
LM plus 1-vessel disease	223 (33.1%)	96 (29.7%)	127 (36.3%)	0.071
LM plus 2-vessel disease	186 (27.6%)	99 (30.7%)	87 (24.9%)	0.093
LM plus 3-vessel disease	81 (12.0%)	46 (14.2%)	35 (10.0%)	0.091
Bifurcation Medina				
1–0–0	129 (19.2%)	53 (16.4%)	76 (21.7%)	0.081
1–0–1	62 (9.2%)	31 (9.6%)	31 (8.9%)	0.740
1–1–0	142 (21.1%)	69 (21.4%)	73 (20.9%)	0.873
1–1–1	116 (17.2%)	59 (18.3%)	57 (16.3%)	0.497

LM—left main, LAD—left anterior descending artery, LCX—left circumflex artery, RCA—right coronary artery, LMCAD—left main coronary artery disease.

**Table 4 jcm-12-00244-t004:** Procedural characteristics.

Variable	Total *n* = 673	Major Troponin Elevation *n* = 323	NO Major Troponin Elevation *n* = 350	*p*-Value (Group 1 vs. Group 2)
PCI success	670 (99.6%)	322 (99.7%)	348 (99.4%)	0.944
Number of stents	1.8 ± 0.9	1.9 ± 0.9	1.7 ± 0.8	<0.001
Total length of implanted stents [mm]	41.1 ± 23.7	45.1 ± 24.9	37.5 ± 21.8	<0.001
Radiation time [min]	18.6 ± 10.0	19.3 ± 10.6	17.8 ± 9.2	0.0957
Radiation dose [mGy]	1410 ± 864	1500 ± 896	1317 ± 822	0.010
Contrast volume [ml]	252 ± 91	257 ± 96	247 ± 87	0.330
Arterial Access site				
Radial	381 (56.6%)	193 (59.8%)	188 (53.7%)	0.114
Femoral	292 (43.4%)	130 (40.2%)	162 (46.3%)	
Stenting LM only	72 (10.7%)	31 (9.6%)	41 (11.7%)	0.375
Stenting LM bifurcation	601 (89.3%)	292 (90.4%)	309 (88.2%)	0.375
One-stent technique	445 (66.1%)	204 (63.2%)	241 (68.9%)	0.119
Two-stents technique	156 (23.2%)	88 (27.2%)	68 (19.4%)	0.016
Two-stents techniques	Total *n* = 156	*n* = 88	*n* = 68	
Crush	60 (38.5%)	34 (38.6%)	26 (38.2%)	0.959
DK-Crush	29 (18.6%)	18 (20.5%)	11 (16.2%)	0.496
Cullote	2 (1.3%)	1 (1.1%)	1 (1.5%)	0.285
T-stenting	25 (16.0%)	12 (13.6%)	13 (19.1%)	0.355
Provisional stenting	40 (25.6%)	23 (26.1%)	17 (25.0%)	0.872
Periprocedural outcomes				
Significant troponin elevation after PCI	323 (48.0%)	323 (100%)	0 (0%)	-
Myocardial Infarction	33 (4.9%)	33 (10.2%)	0 (0%)	-
In-hospital Death	2 (0.3%)	1 (0.3%)	1 (0.3%)	0.425
Stroke	1 (0.1%)	0 (0%)	1 (0.3%)	0.964
Tamponade	2 (0.3%)	1 (0.3%)	1 (0.3%)	0.425
Pulmonary oedema	1 (0.1%)	1 (0.3%)	0 (0%)	0.964
Dissection of aorta	1 (0.1%)	1 (0.3%)	0 (0%)	0.964
Perforation of femoral artery	1 (0.1%)	0 (0%)	1 (0.3%)	0.964
Contrast induced nephropathy	29 (4.3%)	18 (5.6%)	11 (3.1%)	0.121

PCI—percutaneous coronary intervention, LM—left main, DK-crush—double kissing crush technique.

## Data Availability

The data presented in this study are available on request from the corresponding author.

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
