# Peer review of "Peri-Procedural Troponin Elevation after Percutaneous Coronary Intervention for Left Main Coronary Artery Disease"

_jcm, 2022, doi:10.3390/jcm12010244_

Round 1
Reviewer 1 Report
Skorupski et al aimed to investigate the real-world clinical significance of peri procedural troponin elevation after PCI for left main disease. The article is well written. Some minor comments are suggested for the authors.
1. The authors indicate the data is collected from a prospective registry. The nature of the registry is unclear. Ie – who collects the data for this registry? What is the scope of the registry? If previously published, maybe a reference at the first mention of the registry is ideal.
2. A detailed consort diagram of the patient selection may be better in Figure 1. Ie- the number of patients in the registry – the number of subjects excluded and the reasons to be illustrated.
3. Specify either primary objectives and secondary objectives or primary and secondary outcomes of this study in the methods section.
Author Response
Comment 1: The authors indicate the data is collected from a prospective registry. The nature of the registry is unclear. Ie – who collects the data for this registry? What is the scope of the registry? If previously published, maybe a reference at the first mention of the registry is ideal.
Response: We appreciate the reviewer’s comment. The data was collected by the physicians from the 1st Department of Cardiology, Poznan University of Medical Sciences. The study was conducted according to the guidelines of the Declaration of Helsinki and approved by the Institutional Review Board (or Ethics Committee) of Poznan University of Medical Sciences. The registry included all patients with LM coronary artery disease who underwent LM PCI in our department. In this study, we enrolled consecutive 673 patients who underwent LM PCI between January 2015 and February 2021. Patients with significant LM stenosis (≥ 50% diameter) engaging or not the ostial narrowing of left anterior descending artery, circumflex coronary artery or both were enrolled in this study.
We have added following sentences to the manuscript:
“This manuscript is a part of a series of scientific articles on LM disease. The design of the registry have been previously reported [20][21][22].”
Comment 2: A detailed consort diagram of the patient selection may be better in Figure 1. Ie- the number of patients in the registry – the number of subjects excluded and the reasons to be illustrated.
Response: We thank the reviewer for this comment. We followed the recommendations. The edited figure has been added.
Comment 3: Specify either primary objectives and secondary objectives or primary and secondary outcomes of this study in the methods section.
Response: We appreciate the reviewer’s comment. We have added following sentences to the manuscript:
“In-hospital death, in-hospital MI, and long-term all-cause death were established as primary outcome. Secondary outcome of the study was Tn analysis according to various cutoffs of postprocedural Troponin levels.”
Reviewer 2 Report
The prognosis of patients with elevated myocardial enzymes after PCI has always been a concern of cardiologists, and the authors observed the correlation between this marker of myocardial injury and the prognosis after PCI from the relatively important group of patients with left main lesion, and concluded that the mild elevation of this marker did not increase the long-term cardiovascular events and risks of patients, which was of great clinical significance.
Some minor points:
1、please reviewed the most recent articles about the topic if there is any. It is recommended to exclude documents that are too old.
2、The tables and figures might need to be improved: The chart should be independent of the article, so it is necessary to indicate the statistical methods and significant levels used and the statistical results obtained in the remarks of the icon; The icon descriptions in the article are too simplistic to obtain statistical information by looking at them alone; Secondly, the ordinates of the graph are not marked, and the routine needs to be clearly identified.
3、Whether the format of the references in the article should be adjusted to "[4][8][9][12][13][14][15][16][17][18]", which is not very aesthetic, can be simplified as follows: "4,8-9,12-18"
Author Response
Comment 1: Please reviewed the most recent articles about the topic if there is any. It is recommended to exclude documents that are too old.
Response: We appreciate the reviewer’s comment. We have reviewed all recent research on this topic, majority of the articles have been cited and discussed in the “Discussion” section of the manuscript.
Comment 2: The tables and figures might need to be improved: The chart should be independent of the article, so it is necessary to indicate the statistical methods and significant levels used and the statistical results obtained in the remarks of the icon; The icon descriptions in the article are too simplistic to obtain statistical information by looking at them alone; Secondly, the ordinates of the graph are not marked, and the routine needs to be clearly identified.
Response: We sincerely thank the reviewer for this comment. Statistical methods are described in detail in “Materials and Methods” section of the manuscript. We have followed the recommendations. The edited graphs have been added to the manuscript.
Comment 3:. Whether the format of the references in the article should be adjusted to "[4][8][9][12][13][14][15][16][17][18]", which is not very aesthetic, can be simplified as follows: "4,8-9,12-18"
Response: We thank the Reviewer for this comment. Corrected.
Reviewer 3 Report
Dr. Wojciech Jan Skorupski and colleagues used a sample of 323 patients with left main coronary disease who underwent direct stenting to investigate the association of peri-procedural myocardial injury (elevation of TnI) and mortality risk. They found that there was no association between peri-procedural myocardial injury and long-term mortality. Their study was well written and consistent with previous report findings.
I would like to suggest the authors to investigate the association of other cardiac biomarkers, i.e., CPK and CKMB with long-term mortality to highlight the novelty.
Author Response
Comment 1: I would like to suggest the authors to investigate the association of other cardiac biomarkers, i.e., CPK and CKMB with long-term mortality to highlight the novelty.
Response: We thank the reviewer for this relevant comment. We decided to analyze troponin elevations because periprocedural myocardial infarction type 4a (Fourth Universal Definition of MI) is based on troponin levels. However, 48% (323 patients) patients from our group had CK-MB mass level measured before and after LM PCI procedure. As suggested by the reviewer, we investigated the association of CK-MB mass with long-term all-cause mortality.
We have added following sentences to the manuscript:
“3.4. Subanalysis: Different postprocedural CK-MB mass levels cutoffs
48% (323 patients) patients from our group had CK-MB mass level measured before and after LM PCI procedure. Additionally, we decided to investigate the association of CK-MB mass with long-term all-cause mortality in a subanalysis. The Kaplan-Meier curves for all-cause mortality according to different cutoffs of postprocedural CK-MB levels are provided. There is no significant difference in long term mortality (p=0.560) (Figure 5).”
“…(4) a subanalysis of 323 patients showed that the occurrence of CK-MB elevation (> 1x URL; >3x URL; > 5x URL and >10x URL) after LM PCI is also not associated with adverse long-term outcomes.”
“An identical situation, with no effect on long-term death rates, occurs in terms of CK-MB elevations (p=0.560).”
Also, we have added a graph presenting Kaplan-Meier curves for all-cause mortality according to different cutoffs of postprocedural CK-MB mass levels.
Round 2
Reviewer 3 Report
The authors respond to my questions very well.
I have no additional questions.
Author Response
We sincerely thank the Reviewer for valuable and insightful comments.